# Parkin-Mediated Mitophagy by TGF-β Is Connected with Hepatic Stellate Cell Activation

**DOI:** 10.3390/ijms241914826

**Published:** 2023-10-02

**Authors:** Ji Hyun Lee, Kyu Min Kim, Eun Hee Jung, Hye Rim Lee, Ji Hye Yang, Sam Seok Cho, Sung Hwan Ki

**Affiliations:** 1College of Pharmacy, Chosun University, Gwangju 61452, Republic of Koreadmsgmlrmatjd@naver.com (E.H.J.); dgf506@naver.com (H.R.L.); messicho@naver.com (S.S.C.); 2Department of Biomedical Science, College of Natural Science, Chosun University, Gwangju 61452, Republic of Korea; 3College of Korean Medicine, Dongshin University, Naju 58245, Republic of Korea; uranus2k@nate.com

**Keywords:** liver fibrosis, hepatic stellate cell, mitophagy, parkin, Smad2

## Abstract

Hepatic stellate cells (HSCs) are the main contributors to the development and progression of liver fibrosis. Parkin is an E3 ligase involved in mitophagy mediated by lysosomes that maintains mitochondrial homeostasis. Unfortunately, there is little information regarding the regulation of parkin by transforming growth factor-β (TGF-β) and its association with HSC trans-differentiation. This study showed that parkin is upregulated in fibrotic conditions and elucidated the underlying mechanism. Parkin was observed in the cirrhotic region of the patient liver tissues and visualized using immunostaining and immunoblotting of mouse fibrotic liver samples and primary HSCs. The role of parkin-mediated mitophagy in hepatic fibrogenesis was examined using TGF-β-treated LX-2 cells with mitophagy inhibitor, mitochondrial division inhibitor 1. Parkin overexpression and its colocalization with desmin in human tissues were found. Increased parkin in fibrotic liver homogenates of mice was observed. Parkin was expressed more abundantly in HSCs than in hepatocytes and was upregulated under TGF-β. TGF-β-induced parkin was due to Smad3. TGF-β facilitated mitochondrial translocation, leading to mitophagy activation, reversed by mitophagy inhibitor. However, TGF-β did not change mitochondrial function. Mitophagy inhibitor suppressed profibrotic genes and HSC migration mediated by TGF-β. Collectively, parkin-involved mitophagy by TGF-β facilitates HSC activation, suggesting mitophagy may utilize targets for liver fibrosis.

## 1. Introduction

Liver fibrosis is a pathological wound-healing response to acute or chronic hepatic injury caused by hepatitis virus, excess alcohol intake, or metabolic disorders [1]. It is characterized by the accumulation of excessive extracellular matrix (ECM) components, such as type I collagen and fibronectin, which distort the hepatic architecture by forming a fibrous scar and eventually impair organ function. Hepatic stellate cells (HSCs) in the perisinusoidal space are the main source of disproportionate ECM, leading to liver fibrosis [2,3]. HSCs also undergo trans-differentiation from a quiescent phenotype containing lipid droplets to a proliferative myofibroblast phenotype expressing alpha-smooth muscle actin (α-SMA) and increased migratory capacity. However, the signaling pathways involved in HSC activation have not yet been fully elucidated. Thus, the identification of regulatory target molecules and the underlying mechanisms may be crucial for preventing and/or treating liver fibrosis.

In the HSC trans-differentiation process, transforming growth factor-β (TGF-β) is the most powerful profibrogenic cytokine that can mediate the activation of HSCs [4]. The main molecular regulatory mechanism of TGF-β involves combining with the type II TGF-β receptor to phosphorylate the type I TGF-β receptor, which contacts regulatory Smad, such as Smad2 or Smad3. This then translocates into the nucleus along with Smad4 to control the transcription of fibrogenic genes such as collagen 1A1 (col1A1), fibronectin, and plasminogen activator inhibitor-1 (PAI-1) [4,5,6,7].

Aside from the influence of cytokines, other complex environmental factors, such as oxidative stress and energy requirements, can intensify the fibrotic activity of HSCs, which subsequently induces liver fibrosis [8,9]. Mitochondria are intracellular organelles that generate energy and reactive oxygen species (ROS), and mitochondria function as powerhouses for oxidative phosphorylation and control cell survival against ROS-activated signaling cascades [10]. Because mitochondria integrate various cellular signals from the microenvironment and determine the ultimate cell fate, the balance of mitochondrial function is critical for HSC biology [10,11]. Therefore, we hypothesized that distortion of mitochondrial homeostasis may activate HSCs. 

Eradication of damaged mitochondria is of great importance for balancing mitochondrial homeostasis. Cells remove damaged mitochondria through a selective autophagic mechanism known as mitophagy [12]. It mostly refers to mitochondria damaged by depolarization, specifically encapsulated by autophagosomes and fused with lysosomes, followed by degradation of the damaged mitochondria [13]. The main signaling network involved in mitophagy is the PTEN-induced putative kinase 1 (PINK1)-parkin pathway. Although PINK1 is degraded under normal physiological conditions, it translocates from the cytosol to mitochondria upon mitochondrial damage. Mitochondrial depolarization leads to the accumulation of PINK on the outer membrane, which recruits the E3 ubiquitin ligase enzyme parkin via phosphorylation. This leads to parkin activation, which is involved in the degradation of depolarized mitochondria [14]. Several studies have shown that mitophagy plays a crucial role in HSC biology. While studies have shown that mitophagy contributes to the progression of liver fibrosis, contradictory roles have also been documented [15,16,17,18]. These inconsistent outcomes indicate that more effort is required to identify the relationship between mitophagy and HSC activation, eventually leading to liver fibrosis. 

To validate the link between mitophagy and liver fibrosis, we explored the role of mitophagy in HSC activation, which may be mediated by parkin. We found that parkin expression increased and colocalized with desmin in animal and human fibrotic livers. In addition, parkin was more abundant in HSCs than in hepatocytes. The expression and function of parkin in HSCs were validated using in vitro models. We discovered that TGF-β induced parkin expression via Smad3 activation, which promoted mitochondrial sub-localization of parkin and mitophagy. Furthermore, we showed that mitochondrial division inhibitor 1 (Mdivi-1), a mitophagy inhibitor, attenuated hepatic fibrogenesis and HSC migratory capacity. Our study suggests that parkin-mediated mitophagy in HSCs may contribute to the progress of liver fibrosis.

## 2. Results

### 2.1. Upregulation of Parkin in Fibrotic Livers 

First, we examined parkin colocalization with the HSC activation marker desmin in cirrhotic and adjacent normal livers of HCC patients to evaluate the clinical significance of parkin in fibrotic livers. Parkin and desmin were markedly increased and colocalized, indicating parkin upregulation in activated HSCs of fibrotic liver (Figure 1A). Similar changes were observed in the CCl_4_-induced liver fibrosis mice model (Figure 1B). To strengthen our hypothesis regarding the relationship between parkin and liver fibrosis, we examined the parkin levels in fibrotic liver homogenates induced by CCl_4_ treatment or BDL. As expected, parkin expression was increased in fibrotic liver samples compared to their respective controls (Figure 1C,D). PINK1 levels showed different expression patterns in two fibrotic liver samples; i.e., PINK1 expression was increased in liver homogenates of CCl_4_-treated mice but not in liver homogenates of bile duct ligated mice. These results suggest that parkin expression is increased in HSCs under fibrotic conditions. 

### 2.2. Parkin Overexpression In Vitro under Fibrogenic Stimulation

We compared the expression levels of parkin in different types of hepatic cells and found that parkin was more highly expressed in HSCs than in hepatocytes (Figure 2A). Next, we isolated HSCs from mice injected with a single dose of CCl_4_. Parkin expression was enhanced in the HSCs from livers of CCl_4_-injected mice upon HSC activation, representing an increase of α-SMA (Figure 2B). In addition, we treated TGF-β in primary HSCs isolated from mice and found that parkin was increased along with the induction of α-SMA (Figure 2C). This outcome was also observed in LX-2 cells, immortalized human HSCs, at various time points and concentrations. Parkin expression increased after 0.5–12 h of TGF-β treatment and was highest at 12 h (Figure 2D). In addition, parkin expression was gradually induced by TGF-β stimulation up to 2 ng/mL (Figure 2E). These results prove that parkin is induced in HSCs exposed to fibrogenic stimuli during hepatic fibrogenesis.

### 2.3. A Regulatory Mechanism of Parkin Overexpression by TGF-β

To verify whether parkin was regulated at the transcriptional level, we determined the parkin mRNA level in TGF-β-treated LX-2 cells. The mRNA level of parkin was increased at 1–3 h after TGF-β treatment and peaked at 1 h. The parkin mRNA level gradually decreased to 12 h (Figure 3A). Furthermore, we pretreated LX-2 cells with the transcription inhibitor ActD and then exposed the cells to TGF-β. The TGF-β-involved parkin induction was suppressed by pretreatment with ActD (Figure 3B). Hence, these results indicate that TGF-β-mediated parkin expression is transcriptionally regulated. We hypothesize that Smad3, which is a crucial TGF-β signaling pathway mediator [19], may act as a transcription factor for parkin induction. To test this hypothesis, we modulated Smad3 expression using a Smad3 overexpression plasmid. Interestingly, parkin expression was enhanced by ectopic Smad3 expression (Figure 3C), suggesting that parkin expression is controlled in a Smad3-mediated transcriptional manner.

### 2.4. Role of TGF-β Stimulation on Mitophagy

To assess the role of TGF-β-mediated parkin overexpression on liver fibrosis, we investigated the function of TGF-β-induced parkin on mitophagy. We first evaluated parkin localization to the mitochondria upon TGF-β treatment. Parkin translocation to the mitochondria was enhanced in TGF-β-stimulated LX-2 cells, as in CCCP (a well-known mitophagy inducer)-treated cells (Figure 4A). In addition, we investigated the induction of mitophagy after TGF-β incubation based on the expression of mitophagy-related genes. In association with parkin induction by TGF-β, the level of the LC3B-II was increased compared with those of the control (Figure 4B). However, the p62 expression was shown as an irregular pattern after TGF-β treatment. When cells were exposed to Mdivi-1, the increased levels of LC3B-II and parkin by TGF-β were diminished (Figure 4C). This outcome of mitophagy was verified by visualizing mitophagy under TGF-β in the presence of Mdivi-1 (Figure 4D). These data imply that TGF-β induces translocation to the mitochondria and mitophagy in HSCs.

### 2.5. Role of TGF-β Stimulation on Mitochondrial Function 

To investigate the effect of TGF-β on mitochondrial function, we assessed whether TGF-β treatment increased mitochondrial ROS (mtROS) production in LX-2 cells using the mtROS indicator MitoSOX™. Unexpectedly, the mtROS levels did not change after TGF-β stimulation (Figure 5A). However, rotenone, a mitochondrial complex I inhibitor used as a positive control, significantly increased the mtROS levels. A similar effect was seen for the change in mitochondrial membrane potential (MMP) of TGF-β-exposed LX-2 cells based on staining with mitochondria-sensitive Rho-123 (Figure 5B). These results indicate that TGF-β stimulation promotes mitophagy but does not affect mitochondrial function in HSCs.

### 2.6. Impact of TGF-β-Mediated Mitophagy on Liver Fibrogenesis and HSC Migration

To delineate whether TGF-β-mediated mitophagy can affect the profibrotic reaction of the liver, we examined the effect of TGF-β-involved mitophagy on the expression of hepatic fibrogenesis-related genes such as PAI-1, collagen 1A1, and fibronectin. Interestingly, the induction of TGF-β-derived profibrotic genes was suppressed by a chemical mitophagy inhibitor Mdivi-1 treatment (Figure 6A). Consistently, when we check the effect of mitophagy inhibition on HSC migration by wound healing assay, TGF-β-induced cell migration into the wound area was dramatically diminished after Mdivi-1 incubation (Figure 6B). 

Our results corroborated that TGF-β-induced mitophagy is sufficient to facilitate hepatic fibrogenesis and HSC migration, leading to subsequent liver fibrosis. 

## 3. Discussion

Persistent HSC activation causes metabolic stress due to a combination of factors [20]. Mitochondria are intracellular organelles that protect cells against various stress responses through mitochondrial dynamics [21]. Mitophagy has been suggested to be an important mechanism for mitochondrial dynamics, mostly mediated by PINK1/parkin signaling [22]. The fact that the liver is one of the most abundant organs in terms of the number and density of mitochondria underscores the role of mitophagy via controlling mitochondrial homeostasis in liver pathophysiology [23]. Despite the known effects of mitophagy in HSCs [15,16,17,18], the relationship between mitophagy, HSC activation, and subsequent liver fibrosis remains controversial. In this study, we found that parkin is overexpressed in HSCs under fibrotic liver conditions for the first time. Immunoblot and immunostaining data from liver tissues of patients with cirrhosis and mice adopted CCl_4_ showed that parkin upregulation in HSCs may facilitate the progression of liver fibrosis. The immunoblot data from liver homogenates of bile duct ligated mice also support our notion. It might be expected that immunostaining data from the liver section of BDL samples also show the same tendency. Data demonstrating parkin induction in HSCs from CCl_4_-treated mice or TGF-β-treated primary HSCs, as well as LX-2 cells, further supports our findings. In addition, we discovered the significance of parkin in HSC biology based on the outcome of parkin distribution in non-parenchymal HSCs compared with hepatocytes. We did not observe consistent changes in PINK1, another critical protein for mitophagy, in two liver fibrosis-induced samples. Hence, we propose that parkin plays a greater role than PINK1 in HSC activation. This difference in PINK induction in two fibrosis models might be derived from some distinct metabolites which are generated in respective liver fibrosis-inducing reactions; CCl_4_ is a commonly used reagent to induce liver fibrosis by binding to triacylglycerols and phospholipids throughout subcellular fractions and causing lipid peroxidation [24,25]. However, BDL is a surgical method derived from acute obstructive jaundice and does not require the bioactivation of any external toxin as in the CCl_4_ model [26]. 

We examined parkin expression in hepatocytes, a major cell type in the liver. However, parkin expression was not affected by TGF-β treatment in hepatocytes (data not shown). The result from more abundant parkin expression in HSCs compared to in hepatocytes and the number of HSCs incremented with HSC activation during fibrosis, whereas that of hepatocytes declined [27]. These led us to study the role of parkin in HSCs for liver fibrosis. 

The regulatory mechanism of parkin expression has been elucidated; parkin expression is regulated by transcription factors such as p53, activating transcription factor 4, c-jun, and NF-κB and several microRNAs (miRs), including miR-34b/c, miR-218, and miR-146a [28,29,30,31,32,33,34]. We identified Smad3 as a novel parkin transcriptional regulator in the present study. In addition, we also found that post-transcriptional processes, including proteasome or lysosome-dependent protein stability, did not appear to be mechanisms for TGF-β-mediated parkin induction (data not shown). This notion is contrary to the fact that E3 ligases, such as a mitochondrial ubiquitin ligase Nrdp1 and a key protein for lysosomal degradation p32, promote parkin degradation [35,36,37]. Hence, a more in-depth study of parkin regulation by TGF-β is warranted. Further studies are required to validate the precise relationship between Smad3 and parkin induction and a detailed investigation of the other regulatory mechanisms in HSCs.

Parkin is a RING-type E3 ubiquitin ligase controlled by the interaction between the N-terminal Ubl domain and the C-terminus of the protein, and it plays a major role in mitophagy [38]. The fact that basal mitophagy is necessary for intracellular recycling and metabolic regulation via a PINK1/parkin-independent pathway and that parkin-dependent mitophagy is related to the elimination of damaged mitochondria in response to environmental stimuli [39,40] reinforces the importance of parkin-mediated mitophagy during HSC activation. Because mitophagy-involved degradative pathways allow regulation of HSC activation, we hypothesize that mitophagy may control signaling pathways related to phenotypic changes in HSC and ECM accumulation. This notion is supported by a recent report that showed a striking connection between fibrosis and delayed clearance of hepatic cells [41]. However, the mechanism through which mitophagy modulates HSC activation has not been identified yet. Here, we showed that TGF-β induces mitophagy via overexpression of parkin in HSCs. TGF-β treatment allows parkin to translocate into the mitochondria and enhance mitophagy. In addition, we investigated the effect of TGF-β on mitochondrial function. Based on the results from no effect of TGF-β on mtROS production or the MMP in HSCs, we assumed that TGF-β treatment does not change calcium uptake and ATP synthesis, which were potentially influenced by MMP [42,43,44]. We thus conclude that parkin-mediated mitophagy might contribute to the balance of mitochondrial dynamics via clearance of TGF-β-induced defective mitochondria [45]. Further studies are necessary to explain this phenomenon.

Our finding that TGF-β-mediated mitophagy promoted the induction of profibrotic genes suggests the functional role of mitophagy in HSC during liver fibrosis. This outcome is strengthened by our result that mitophagy inhibitor Mdivi-1 treatment attenuated increased expression of fibrogenic genes. Moreover, we observed the migratory effect of mitophagy in HSCs. Despite the reports on the conflicting expression and role of parkin-mediated mitophagy [46,47,48], our findings demonstrated that these profibrotic effects of mitophagy might be associated with parkin induction. This discrepancy between results may be derived from cell variation, optimized conditions, and/or involvement in growth conditions. Moreover, the expression and the role of parkin were also studied in hepatocellular carcinoma, which is an advanced disease from liver fibrosis. While we proposed parkin as a profibrotic molecule for liver fibrosis, it was revealed that parkin deficiency induced hepatocyte proliferation and resistance to apoptosis, resulting in the development of hepatocellular carcinoma [49,50]. 

In conclusion, this study demonstrated that parkin is upregulated by TGF-β stimulation and elicited by Smad3 in HSCs. Parkin induction activates mitophagy, which may affect liver fibrogenesis and HSC migration, which results in liver fibrosis (Figure 7). These findings suggest that parkin may act as a therapeutic target for liver fibrosis.

## 4. Materials and Methods

### 4.1. Materials

Antibodies against parkin and Smad2/3 were purchased from Cell Signaling Technology (Beverly, MA, USA). Antibodies against PAI-1, p62, and fibronectin were obtained from BD Biosciences (Mountain View, CA, USA). Anti-LC3B and anti-PINK1 antibodies were provided by NOVUS Biologicals (Littleton, CO, USA). Anti-desmin and anti-parkin antibodies for immunofluorescence and anti-col1A1 antibody were purchased from Abcam (Cambridge, MA, USA). Horseradish peroxidase-conjugated goat anti-rabbit and anti-mouse antibodies, MitoSOX™, and MitoTracker™ was purchased from Invitrogen (Carlsbad, CA, USA). Rhodamine123 (Rho-123) was purchased from Santa Cruz Biotechnology (Dallas, TX, USA). Antibodies against albumin were provided by CUSABIO (Houston, TX, USA). Antibodies against α-SMA and β-actin, actinomycin-D (ActD), mitophagy inhibitor mitochondrial division inhibitor 1 (Mdivi-1), carbonyl cyanide *m*-chlorophenyl hydrazone (CCCP), and rotenone were purchased from Sigma-Aldrich (St. Louis, MO, USA). Recombinant human TGF-β (Cat no. 240-B) was obtained from R&D Systems (Minneapolis, MI, USA).

### 4.2. Cell Culture

LX-2 cells (immortalized activated human HSCs) were provided by Dr. S. L. Friedmann (Mount Sinai School of Medicine, NY, USA). The cells were plated in 60 mm plates at 1 × 10^5^ cells/per well and grown to 70–80% confluence. The cells were maintained in DMEM containing 10% FBS (HyClone, Logan, UT, USA) and 50 U/mL penicillin/streptomycin at 37 °C in a humidified 5% CO_2_ atmosphere. The cells were washed with cold PBS before sample preparation.

### 4.3. Primary HSCs and Hepatocytes Isolation 

Hepatocytes and HSCs were isolated from 8-week-old mice (Oriental Bio, Sungnam, Republic of Korea) as previously reported [51]. After intubation in the portal vein, the livers were perfused in situ with Ca^2+^-free Hank’s balanced saline solution at 37 °C for 15 min and then perfused with the solution containing 0.05% collagenase and Ca^2+^ for 15 min at a flow of 10 mL/min. The perfused livers were minced, filtered through a 70 μm cell strainer (BD Bioscience), and centrifuged at 50× *g* for 3 min to generate a supernatant and pellet. The pelleted hepatocytes were resuspended in DMEM supplemented with 10% FBS, 100 U/mL penicillin, 100 μg/mL streptomycin, 5 mM HEPES, and 10 nM dexamethasone. HSCs were isolated as previously described. Briefly, the supernatant was centrifuged at 500× *g* for 10 min, resuspended in Ficoll^®^ plus Percoll^®^ (1:10; GE Healthcare, Chicago, IL, USA), and centrifuged at 1400× *g* for 15 min. The HSCs were collected from the interface. Isolated primary HSCs were seeded in uncoated dishes and harvested.

### 4.4. Experimental Animals and Ethics Statement

Male ICR mice were obtained from Oriental Bio (Sungnam, Republic of Korea). The Institutional Ethics Committee for Animal Experimentation approved this study. All experiments and postoperative animal care procedures were performed in accordance with the governmental and international guidelines on animal experimentation. CCl_4−_ or bile duct ligation (BDL)-induced fibrosis mouse samples were adopted as previously used [52]. CCl_4_ (0.5 mg/kg; dissolved in 10% olive oil) was intraperitoneally administered to mice three times a week for two weeks. For BDL, the common bile duct was exposed near its attachment to the duodenum and ligated with 6.0 silk. In the case of sham-operated mice, the bile duct was dissected and similarly manipulated but not ligated. 

### 4.5. Patient Samples 

Human fibrotic or non-fibrotic liver samples obtained from ten cancer patients were separated by histological examination and ultrasonography at Chosun University Hospital in South Korea, as described in a previous study [53]. The study protocol was approved by the Institutional Review Board of Chosun Medical Center (#2013-04-005). The study was consistent with the Declaration of Helsinki. Informed consent was obtained from human subjects, and the subject’s right to privacy was observed.

### 4.6. Immunoblot Analysis 

Protein extraction, SDS polyacrylamide gel electrophoresis, and immunoblot analyses were performed as previously described [54]. Briefly, cell lysates were separated by electrophoresis on 7.5 or 12% gels and then electrophoretically transferred to nitrocellulose membranes (GE Healthcare, Chicago, IL, USA). The nitrocellulose membranes were incubated with the indicated primary antibody and then with a horseradish peroxidase-conjugated secondary antibody (Invitrogen). Immunoreactive proteins were visualized using an enhanced chemiluminescence detection kit (Amersham Biosciences, Upsalla, Sweden). Equal protein loading was verified based on the β-actin or GAPDH level. Scanning densitometry was done using an Image J (National Institutes of Health, Bethesda, MD, USA). At least three samples were used for each experiment. 

### 4.7. RNA Isolation and RT-PCR Analysis 

Total RNA was extracted using TRIzol^®^ reagent (Invitrogen, Carlsbad, CA, USA) according to the manufacturer’s instructions. RNA (2 µg) was reverse transcribed using oligo (dT)_16_ primers to obtain cDNA. The cDNA was amplified using a high-capacity cDNA synthesis kit (Bioneer, Daejeon, Republic of Korea) with a thermal cycler (Bio-Rad, Hercules, CA, USA). The amplified products were separated on a 2% agarose gel, stained with ethidium bromide (Sigma-Aldrich, St. Louis, MO, USA), and visualized with a gel documentation system (Fujifilm, Tokyo, Japan). The primer sequences were as follows: human parkin sense 5′-TCCTTCCTGCTGTCAGTGTG-3′, and antisense 5′-GCAGAGACCGTGGAGAAAAG-3′; human GAPDH sense 5′-GAAGGTGAAGGTCGGAGTC-3′, and antisense 5′-GAAGATGGTGATGGGATTTC-3′. Equal loading was verified based on the GAPDH level.

### 4.8. Transient Transfection 

The plasmid pCDNA3-Flag-Smad3 was kindly provided by Prof. H. S. Choi (Chonnam National University, Gwangju, Republic of Korea) [55]. The plasmid pRK5-Myc-Parkin was purchased from Addgene (Cambridge, MA, USA). LX-2 cells were transfected with Smad3 or MOCK plasmid in the presence of Lipofectamine^TM^ 2000 (Invitrogen) for 24 h. The transfected cells were allowed to recover in MEM with 1% FBS overnight.

### 4.9. Confocal Microscopy 

LX-2 cells were treated with 2 ng/mL TGF-β and 5 μM CCCP (positive control for mitophagy) for 6 h. To identify depolarized mitochondria, the cells were simultaneously stained with 200 nM MitoTracker™ (Invitrogen) for 30 min at 37 °C in a humidified 5% CO_2_ atmosphere. The treated cells were fixed with 4% paraformaldehyde, followed by permeabilization with 0.1% Triton X-100. The samples were immunostained with antibodies directed against parkin for 1 h, followed by incubation with Alexa Fluor^®^ 555 goat anti-rabbit IgG (Invitrogen). To detect mitophagy, LX-2 cells were infected with adenoviral GFP-LC3B and treated with TGF-β in the presence or absence of 10 μM Mdivi-1. MitoTracker™ was added for 30 min before fixation with paraformaldehyde for 30 min. After incubation, the samples were covered with a mounting medium and a coverslip. The cell samples were immunostained with antibodies directed against E6AP overnight, followed by incubation with Alexa Fluor^®^ 488 goat anti-rabbit IgG (Invitrogen). Tissue sections were deparaffinized and incubated with antibodies of parkin and desmin, at 37 °C, overnight and 4 h, respectively, followed by incubation with Alexa Fluor^®^ 594 goat anti-rabbit IgG (Biolegend, San Diego, CA, USA) or Alexa Fluor^®^ 488 goat anti-rabbit IgG (Biolegend, San Diego, CA, USA) at 37 °C for 3 h. After incubation, the samples were cover-slipped with mounting media. The samples were examined using a laser scanning confocal microscope (A1; Nikon Instruments Inc., Melville, NY, USA). Photographs of the sections with orange-yellow dots with a given intensity value were segmented. 

### 4.10. Mitochondrial Membrane Potential (MMP) Assay

LX-2 cells were treated with or without 2 ng/mL TGF-β, or 10 μM rotenone as a positive control, at 37 °C for 18 h and then incubated with 0.05 ng/mL Rho-123 at 37 °C for 30 min. The cells were harvested by trypsinization and washed with PBS. The Rho-123 fluorescence (λex/em = 396/610 nm) was detected using channel FL1-A of a flow cytometry (Beckman-Coulter, Kristiansand, Norway).

### 4.11. Measurement of Mitochondrial ROS (mtROS) Levels

LX-2 cells were treated with or without 2 ng/mL TGF-β at 37 °C for 15 min, then cells were incubated with 10 μM MitoSOX™ at 37 °C for 30 min. 10 µM Rotenone was used as a positive control. The cells were harvested by trypsinization and washed with PBS. The MitoSOX™ fluorescence was detected using channel FL2-A of flow cytometry (Beckman-Coulter, Kristiansand, Norway).

### 4.12. Wound Healing Assay

LX-2 cells were seeded at a density of 3 × 10^5^ per well in culture inserts (ibidi Culture-Insert 4 wells, ibidi GmbH, Gräfelfing, Germany). Culture inserts were removed when the confluency of cells reached 80–90%. Then, cells were washed with PBS to remove floating cells, and the medium was changed to MEM media containing 0.1% FBS. Cells were treated with 10 μM Mdivi-1 in the presence or absence of 2 ng/mL TGF-β for 12 h, and photomicrographs were taken at 0 and 12 h at 50× magnification using a camera attached to a microscope (Axiovert 200 M, Carl ZEISS, Germany).

### 4.13. Statistical Analysis

Student *t*-test was performed to assess the significance of differences among treatment groups. The data were expressed as means ± SE, and the criterion for statistical significance was set at *p* < 0.05 or *p* < 0.01.

## Figures and Tables

**Figure 1 ijms-24-14826-f001:**
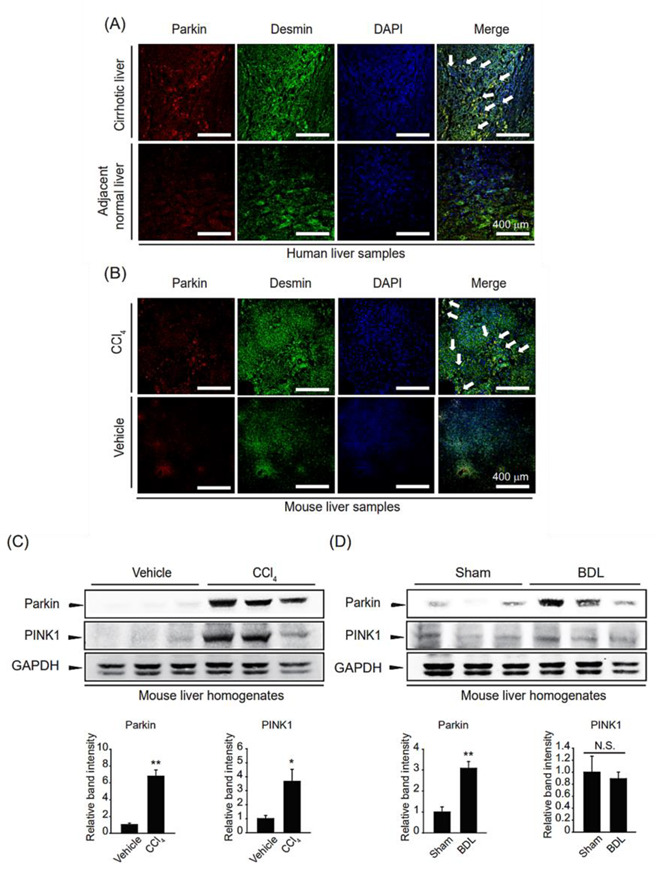
Parkin overexpression in fibrotic liver samples. (**A**) Parkin and desmin immunostaining in cirrhotic patient and adjacent normal liver samples (magnification: 200×, Scale bar = 400 µm). White arrows indicate the colocalization of parkin and desmin. (**B**) Parkin and desmin immunostaining in carbon tetrachloride (CCl_4_)-injected mice liver sections (magnification: 200×, Scale bar = 400 µm). White arrows indicate the colocalization of parkin and desmin. (**C**,**D**) Immunoblotting for parkin and PINK1 of liver samples from mice with fibrosis. For (**C**), the mice were injected with CCl_4_ for two weeks. For (**D**), the mice underwent bile duct ligation (BDL). Protein levels of parkin or PINK1 were assessed by immunoblot analysis. GAPDH was used to assess equal protein loading. Parkin level was assessed by scanning densitometry. The data represents the mean ± standard error (SE) (*n* = 3, significant difference versus respective controls, * *p* < 0.05, ** *p* < 0.01, N.S. not significant).

**Figure 2 ijms-24-14826-f002:**
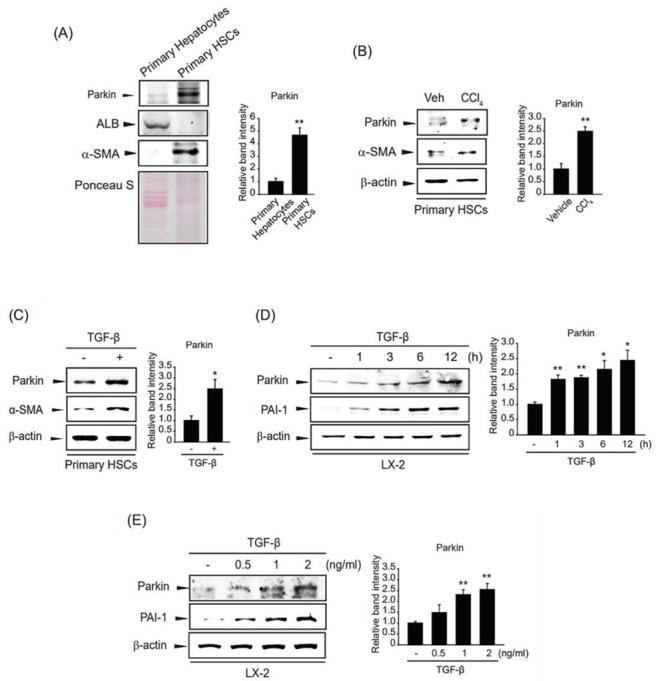
Parkin upregulation upon TGF-β stimulation of hepatic fibrogenesis in HSCs. (**A**) Parkin expression in mouse primary hepatocytes and hepatic stellate cells (HSCs). Immunoblotting was performed on the cell lysates. β-actin was used to verify equal loading of proteins. Albumin (ALB) and α-smooth muscle actin (α-SMA) were detected as markers of hepatocytes and HSCs, respectively. Ponceau S staining represents equal loading of proteins. Parkin level was assessed by scanning densitometry. The data represents the mean ± standard error (SE) (*n* = 3, significant difference versus primary hepatocytes, ** *p* < 0.01). (**B**) Expression of parkin in primary HSCs from livers of mice administered a single dose (0.5 mg/kg) of CCl_4_ for 24 h. The level of parkin or α-SMA was assessed by immunoblotting. Parkin level was assessed by scanning densitometry. The data represents the mean ± standard error (SE) (*n* = 3, significant different versus primary HSCs isolated from vehicle-injected mice, ** *p* < 0.01). (**C**) Effect of transforming growth factor-β (TGF-β) on parkin expression in primary HSCs. Primary HSCs were isolated and treated with 2 ng/mL TGF-β for 12 h. The expression of parkin or α-SMA was evaluated by immunoblotting. Parkin level was assessed by scanning densitometry. The data represents the mean ± standard error (SE) (*n* = 3, significant different versus vehicle-treated primary HSCs, * *p* < 0.05). (**D**) The time courses of parkin expression in TGF-β-treated LX-2 cells. Parkin protein was immunoblotted in the lysates of cells incubated with 2 ng/mL TGF-β for 0.5–24 h. Parkin level was assessed by scanning densitometry. The data represents the mean ± standard error (SE) (*n* = 3, significant different versus control, * *p* < 0.05, ** *p* < 0.01). (**E**) The effect of various concentrations of TGF-β on parkin upregulation in LX-2 cells. Parkin was detected in the lysates of cells incubated with 0.5–2 ng/mL TGF-β for 12 h. Parkin level was assessed by scanning densitometry. The data represents the mean ± standard error (SE) (*n* = 3, significant different versus control, ** *p* < 0.01).

**Figure 3 ijms-24-14826-f003:**
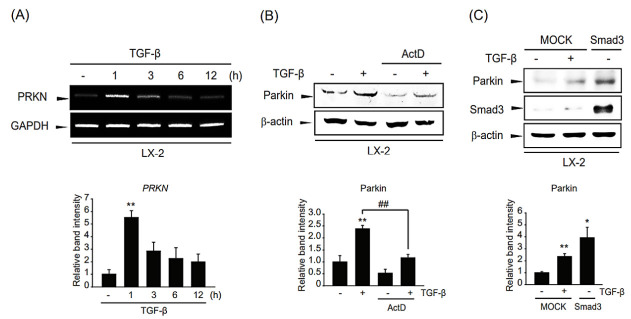
Mechanistic study of the regulation of parkin induction by TGF-β. (**A**) RT-PCR analysis. LX-2 cells were treated with 2 ng/mL TGF-β for 1–12 h. The expression level of parkin mRNA was assessed by RT-PCR using *GAPDH* as an internal control. Parkin level was assessed by scanning densitometry. The data represents the mean ± standard error (SE) (*n* = 3, significant different versus control, ** *p* < 0.01). (**B**) Effect of actinomycin D (ActD) on parkin induction by TGF-β in LX-2 cells. The cells were treated with 5 μg/mL of ActD with or without TGF-β treatment. The level of parkin was determined after 1 ng/mL TGF-β treatment for 12 h. Parkin level was assessed by scanning densitometry. The data represents the mean ± standard error (SE) (*n* = 3, significant different versus respective controls, ** *p* < 0.01; significant versus TGF-β-treated cells, ## *p* < 0.01). (**C**) Involvement of Smad3 in TGF-β-induced parkin expression. LX-2 cells were transfected with pcDNA3.1 (MOCK) or Smad3 for 24 h. MOCK overexpressed cells were treated with 1 ng/mL TGF-β for 6 h. Parkin level was assessed by scanning densitometry. The data represents the mean ± standard error (SE) (*n* = 3, significant different versus MOCK-transfected cells, * *p* < 0.05, ** *p* < 0.01).

**Figure 4 ijms-24-14826-f004:**
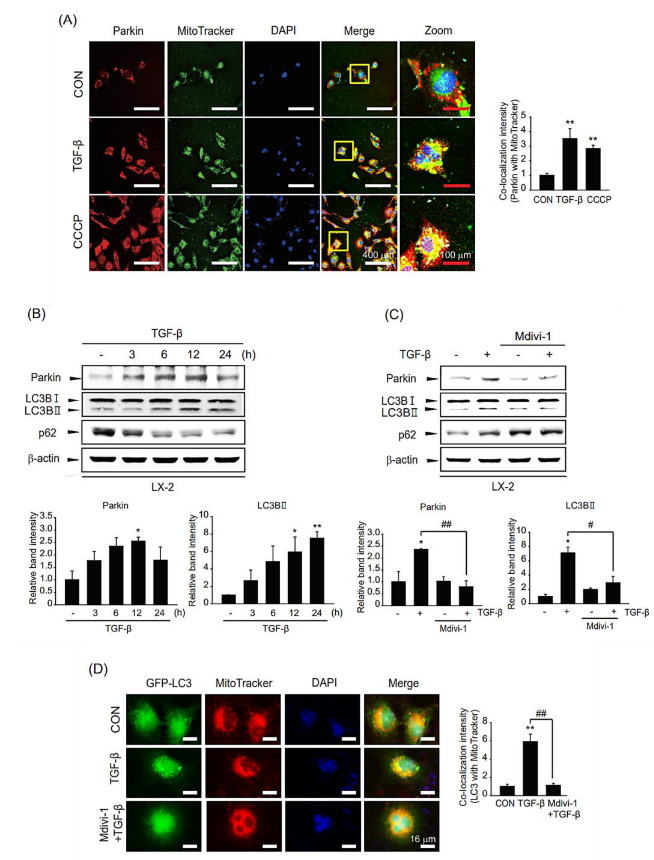
Role of TGF-β-induced parkin on mitophagy in LX-2 cells. (**A**) Immunostaining for parkin in TGF-β-stimulated LX-2 cells and their quantification. After treatment with 2 ng/mL TGF-β for 18 h, the cells were stained with MitoTracker™ Green (200 nM). CCCP was utilized as a positive control. Red: parkin, green: mitochondria, blue: DAPI, orange-yellow: merge. (white scale bar = 400 µm, red scale bar = 100 µm). The colocalization intensity of parkin with MitoTracker was evaluated by scanning densitometry. Ten different cells were randomly selected for each sample. The data represents the mean ± standard error (SE). (*n* = 3, significant different versus control, ** *p* < 0.01). (**B**) The effect of TGF-β on mitophagic activity. Level of parkin, LC3B, or p62 in LX-2 cells treated with 2 ng/mL TGF-β for various times (3–24 h) was detected by immunoblotting. Parkin or LC3BII level was assessed by scanning densitometry. The data represents the mean ± standard error (SE) (*n* = 3, significant different versus control, * *p* < 0.05, ** *p* < 0.01). (**C**) Effect of mitophagy inhibitor Mdivi-1 on TGF-β-mediated mitophagy. Expression of LC3B or p62 was measured after treatment with 2 ng/mL TGF-β sequential to 10 μM Mdivi-1 treatment. Parkin or LC3BII level was assessed by scanning densitometry. The data represents the mean ± standard error (SE) (*n* = 3, significant different versus control, * *p* < 0.05; significant versus TGF-β-treated cells, # *p* < 0.05, ## *p* < 0.01). (**D**) Representative image of mitophagy (**left**) and their quantification (**right**). LX-2 cells were infected with adeno-LC3B and then treated with MitoTracker™ (200 nM). Subsequently, cells were incubated with 2 ng/mL TGF-β for 12 h in the presence or absence of 10 μM Mdivi-1. Green: LC3B, red: MitoTracker™, blue: DAPI, orange-yellow: merge. Orange-yellow puncta represent mitochondrial digestion by autophagy (Scale bar = 16 µm). The colocalization intensity was evaluated by scanning densitometry. Ten different cells were randomly selected for each sample. The data represents the mean ± standard error (SE). (*n* = 3, significant different versus control, ** *p* < 0.01, significant versus TGF-β-treated cells, ## *p* < 0.01).

**Figure 5 ijms-24-14826-f005:**
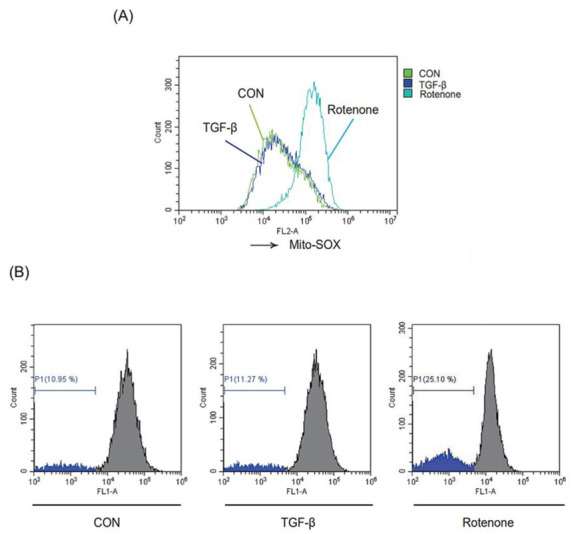
Effect of TGF-β exposure on mitochondrial function. (**A**) Effect of TGF-β on mitochondrial ROS. LX-2 cells were treated with or without 2 ng/mL TGF-β for 15 min, or 10 μM rotenone for 1 h, and then incubated with 10 μM MitoSOX™ for 30 min. Rotenone was used as a positive control. The cells were analyzed by flow cytometry (green: control, blue: TGF-β, sky-blue: rotenone). (**B**) Effect of TGF-β on mitochondrial membrane potential (MMP). MMP was measured by rhodamine-123 (Rho-123) staining and analyzed by flow cytometry. LX-2 cells were exposed to 2 ng/mL TGF-β or 10 μM rotenone for 18 h and then sequentially loaded with 0.05 ng/mL Rho-123 for 30 min. The inserted histogram exhibits a left shift of the histogram peak, illustrating the decrease of Rho-123 fluorescence intensity due to the loss of MMP. The percentage of cells with reduced fluorescence intensity was calculated.

**Figure 6 ijms-24-14826-f006:**
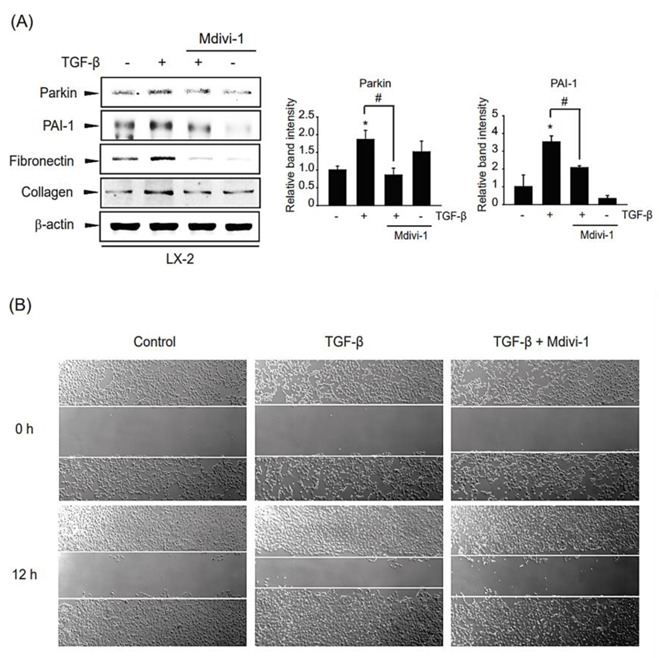
Role of TGF-β-induced mitophagy on liver fibrogenesis and HSC migration in LX-2 cell (**A**) Effect of Mdivi-1 on hepatic fibrogenesis. LX-2 cells were pretreated with 10 μM Mdivi-1 for 0.5 h, and then were incubated with 1 ng/mL TGF-β for 12 h, and then liver fibrogenesis-related gene expression in the cell lysates was detected by immunoblotting. Parkin or PAI-1 level was assessed by scanning densitometry. The data represents the mean ± standard error (SE) (*n* = 3, significant different versus respective controls, * *p* < 0.05; significant versus TGF-β-treated cells, # *p* < 0.05). (**B**) Effect of Mdvi-1 on TGF-β-derived cellular migration by wound healing assay. LX-2 cells were treated with 10 µM Mdvi-1 with or without stimulation of TGF-β (2 ng/mL) after creating wounds.

**Figure 7 ijms-24-14826-f007:**
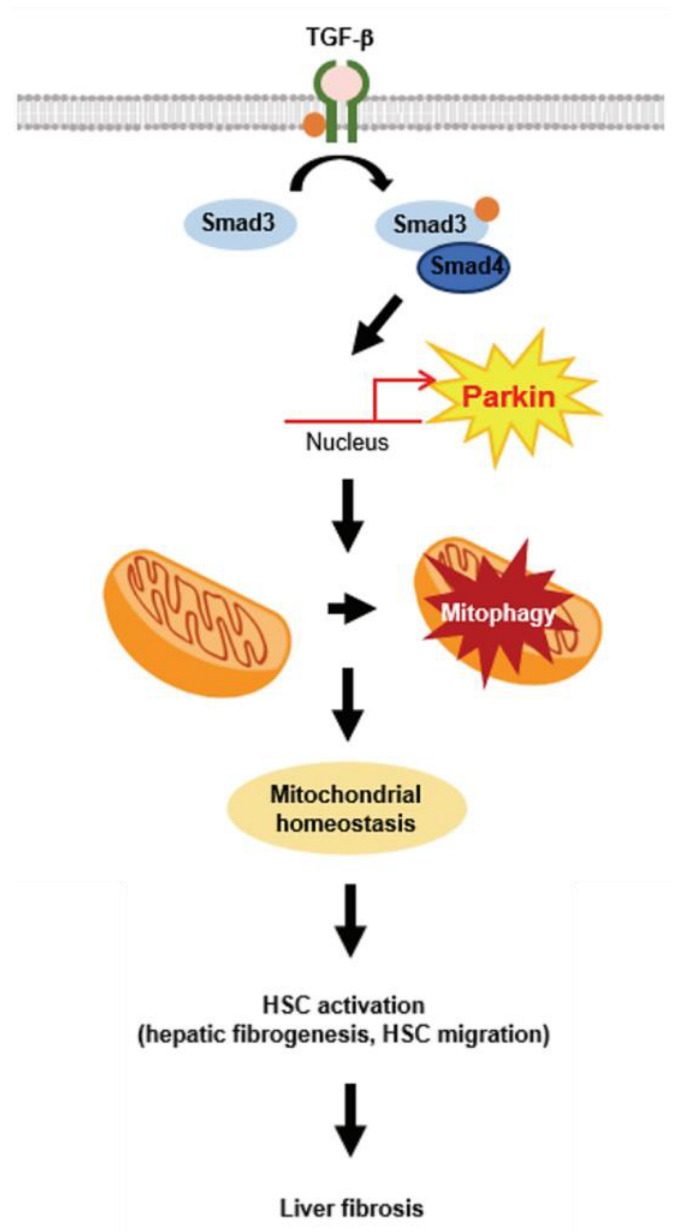
Schematic illustration. Schematic diagram of the mechanism by which TGF-β-mediated mitophagy promotes profibrotic effect via parkin.

## Data Availability

The data presented in this study are available in the article.

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
