# Peer review of "Parkin-Mediated Mitophagy by TGF-β Is Connected with Hepatic Stellate Cell Activation"

_ijms, 2023, doi:10.3390/ijms241914826_

Round 1

Reviewer 1 Report

This manuscript showed that TGF-b/parkin-mediated mitophagy might play a role in HSC activation using LX2, an HSC cell line, primary HSCs from mice received short-term CCl4 and BDL, and also livers from HCC patients. Although the organization and rational of each section is fairy great, the presentation of results and result interpretations are needed to be improved extensively. Here are my suggestions.

- The author always stated that the treatment increase or decrease the proteins or mRNA. However, only one representative figure (qualitative data of staining, blot, gel) of each experiment is showed and most of them are difficult to be interpreted by eyes. Therefore, I would highly recommend making semi-quantitative data and reveal how many samples/representations (passages, livers, mice) were performed for each experiment.    

- The introduction part is generally fine, except for the last paragraph which is the conclusion of this study.

- The author found that PINK1 expression was increased in liver homogenates of CCl4-treated mice, but not in liver homogenates of BDL mice. However, the signaling of PINK1 is closely dependent on parkin as described in the introduction part. Therefore, there should be other mediators to activate parkin? 

- Following the previous comment, why don't the authors reveal the expression of perkin in BDL using fluorescent staining?

- In fig 1A and B, I cannot see any perkin (red) in merged figure. So the colocalization of desmin and perkin is unclear.

- It is impossible to induce liver fibrosis by injecting CCl4 only for 2 wks or a single dosage. This short administration of CCl4 can trigger only inflammatory responses. Also, there is no clear evidence of established fibrosis, excessive deposition of extracellular matrix, in both fibrosis models. Therefore, the author should not extrapolate that the finding would play a major role in fibrogenesis, but it could be associated with HSC activation merely.

- Following the previous comment, the protocol/dosage for CCl4 injection and BDL should be clearly stated in the manuscript.

- According to the blot shown in figure 2C, I don't think the author could conclude that parkin was increased along with the induction of PAI-1 and fibronectin. a-SMA may be slightly higher. 

- What is the type and source of TGF-b? Did the authors use BSA/HSA to reduce its binding to unwanted surface.

- The treatment period and concentration of TGF-b in LX-2 are inconsistency in all experiments. In fig 2D. 12h and 2ng/mL are the highest time point/conc. Then 6h and 1ng/mL are used in fig 3C. Also 18h and 15mins in fig 4 and 5. What is the reason for this inconsistency? How the reader know that the result would be correct in this case?

- Since the action of Smad3 require phosphorylation, therefore, phosphorylated-Smad3 would be a better marker than total Smad3 as shown in fig 3C. Please clarify why the amount of Smad3 would be sufficient for this study.

- I'm not sure if this journal still accepts conventional RT-PCR gel result as shown in fig3A to indicate the amount of mRNA, since most of international journals require at least quantitative PCR (pPCR).

- The blot of parkin in fig 4C cannot be seen.

- In fig 4B, parkin is highest at 24h, not 12h as shown in fig 2D. Therefore, the prior conclusion was not correct.

- In fig 7, to the best of my knowledge, Smad2 or Smad3 form a homodimer to bind with Smad4 and navigate to the nucleus of HSCs, not a heterodimer between Smad2 and Smad3. Please verify this information.

- This study used the livers from HCC patients as a cirrhotic sample. Is it possible that parkin is upregulated in HCC irrespective of fibrosis?

-

Author Response

Reviewer #1 This manuscript showed that TGF-b/parkin-mediated mitophagy might play a role in HSC activation using LX-2, an HSC cell line, primary HSCs from mice received short-term CCl4 and BDL, and also livers from HCC patients. Although the organization and rational of each section is fairy great, the presentation of results and result interpretations are needed to be improved extensively. Here are my suggestions.

  1. The author always stated that the treatment increase or decrease the proteins or mRNA. However, only one representative figure (qualitative data of staining, blot, gel) of each experiment is showed and most of them are difficult to be interpreted by eyes. Therefore, I would highly recommend making semi-quantitative data and reveal how many samples/representations (passages, livers, mice) were performed for each experiment.

Answer: As the reviewers suggested, we have included semi-quantitative analysis data and information on samples/representations in the revised manuscript.

  1. The introduction part is generally fine, except for the last paragraph which is the conclusion of this study.

Answer: We have revised the last paragraph of introduction as the reviewers pointed out.

  1. The author found that PINK1 expression was increased in liver homogenates of CCl4-treated mice, but not in liver homogenates of BDL mice. However, the signaling of PINK1 is closely dependent on parkin as described in the introduction part. Therefore, there should be other mediators to activate parkin?

Answer: CCl4 is a most commonly used laboratory reagent marked by its toxicity leading to liver inflammation and fibrosis [1]. It binds to triacylglycerols and phospholipids throughout subcellular fractions and causes lipid peroxidation [2]. However, BDL is a surgical method used to induce liver fibrosis, which is derived from an acute obstructive jaundice [3]. In the BDL model, no bioactivation of any external toxin is required as it occurs in the CCl4 model. Hence, we imply that the difference in PINK1 induction in two models derived from some distinct metabolites which are generated in respective liver fibrosis animal model. This was mentioned in discussion of the revised manuscript.

  1. Following the previous comment, why don't the authors reveal the expression of parkin in BDL using fluorescent staining?

Answer: As the reviewer recommended, we tried to obtain bile duct ligated mice liver sample for immunofluorescent to assess parkin overexpression. Unfortunately, we could not get samples. Instead, we imply and the possibility of parkin overexpression in HSCs of BDL samples in the revised manuscript. Authors hope that our efforts would satisfy the reviewer’s point.      

  1. In fig 1A and B, I cannot see any parkin (red) in merged figure. So the colocalization of desmin and perkin is unclear.

Answer: To demonstrate the parkin expression in HSCs, we performed double staining with parkin and desmin specific antibodies, respectively. Green fluorescence represents desmin, which is a specific marker for HSC activation, and red fluorescence indicates parkin expression. Thus, parkin expression in HSC was shown as yellow fluorescence. Our previous reports utilized this method to prove the protein expression in HSCs [4, 5].

  1. It is impossible to induce liver fibrosis by injecting CCl4 only for 2 wks or a single dosage. This short administration of CCl4 can trigger only inflammatory responses. Also, there is no clear evidence of established fibrosis, excessive deposition of extracellular matrix, in both fibrosis models. Therefore, the author should not extrapolate that the finding would play a major role in fibrogenesis, but it could be associated with HSC activation merely.

Answer: Authors thank the reviewer for meticulous reading of our manuscript. We previously reported CCl4 administration for 2 weeks induce HSC activation and liver fiborosis [6]. Moreover, our result from primary HSCs isolated from single CCl4-injected mice also support the fibrogenic effect of CCl4 in Fig. 2B of the revised manuscript.    

  1. Following the previous comment, the protocol/dosage for CCl4 injection and BDL should be clearly stated in the manuscript.

Answer: As the reviewer suggested, we stated the protocol/dosage for CCl4 injection and BDL in detail in the revised manuscript.

  1. According to the blot shown in figure 2C, I don't think the author could conclude that parkin was increased along with the induction of PAI-1 and fibronectin. a-SMA may be slightly higher.

Answer: We agreed reviewer’s comment. Fig. 2C of the revised manuscript was rearranged to conclude parkin overexpression along with HSC activation.

  1. What is the type and source of TGF-b? Did the authors use BSA/HSA to reduce its binding to unwanted surface.

Answer: Recombinant human TGF-b1 protein supplied by R&D systems (Cat no. 240-B) was utilized in our study.

  1. The treatment period and concentration of TGF-b in LX-2 are inconsistency in all experiments. In fig 2D. 12 h and 2ng/mL are the highest time point/conc. Then 6 h and 1 ng/mL are used in fig 3C. Also 18 h and 15 mins in fig 4 and 5. What is the reason for this inconsistency? How the reader know that the result would be correct in this case?

Answer: Several experiments were carried out to prove the role of parkin by TGF-b. We optimized the best conditions to demonstrate our hypothesis as previously reported [4, 5, 7].

  1. Since the action of Smad3 require phosphorylation, therefore, phosphorylated-Smad3 would be a better marker than total Smad3 as shown in fig 3C. Please clarify why the amount of Smad3 would be sufficient for this study.

Answer: We hypothesize that Smad3, which is a crucial TGF-b signaling pathway mediator [8], may act as a transcription factor for parkin induction. To test this hypothesis, we modulated Smad3 expression using a Smad3 overexpression plasmid and found that parkin expression was enhanced by ectopic Smad3 expression. Hence, we examined total Smad3 expression to check Smad3 overexpression as a control.

  1. I'm not sure if this journal still accepts conventional RT-PCR gel result as shown in fig 3A to indicate the amount of mRNA, since most of international journals require at least quantitative PCR (qPCR).

Answer: As the reviewer’s point, the result of conventional PCR was quantified, and the relative changes were statistically analyzed as shown in Fig. 3A of the revised manuscript to reinforce the accuracy and reproducibility of our experiments.    

  1. The blot of parkin in fig 4C cannot be seen.

Answer: The blot of parkin in Fig. 4C was replaced with better one.

  1. In fig 4B, parkin is highest at 24h, not 12h as shown in fig 2D. Therefore, the prior conclusion was not correct.

Answer: The Fig. 2D was replaced with another blot. Additionally, the band intensity was quantified, and the relative changes were statistically analyzed as shown in Fig. 2D and 2E of the revised manuscript.    

  1. In fig 7, to the best of my knowledge, Smad2 or Smad3 form a homodimer to bind with Smad4 and navigate to the nucleus of HSCs, not a heterodimer between Smad2 and Smad3. Please verify this information.

Answer: The authors thank the reviewer for the helpful comment. As the reviewer suggested, we modified Fig.7 of the revised manuscript.

  1. This study used the livers from HCC patients as a cirrhotic sample. Is it possible that parkin is upregulated in HCC irrespective of fibrosis?

Answer: Although we used cirrhotic and adjacent normal liver sample from HCC patients, we did not examine parkin expression in respect to HCC. Instead, we discussed some reports about the expression and the role of parkin in HCC in the revised manuscript [9, 10].    

Reviewer 2 Report

In this manuscript the authors describe the upregulation of the protein Parkin during TGF-b-mediated hepatic stellate cell (HSC) activation and investigate the molecular mechanisms. The results suggest an involvement of mitophagy during HSC activation. Paradoxically, TGF-b did not affect mitochondrial function but mitophagy inhibitors did affect HSC activation.

The authors show that Parkin is expressed both in hepatocytes and in HSCs. This manuscript does not address this but focusses on its expression in HSCs. However, on several occasions ‘HSC activation’ and ‘liver fibrosis’ seem to be used almost interchangeably. For example: i) the last sentence of the Abstract: since Parkin-mediated mitophagy is also likely involved in hepatocyte function, targeting parkin in HSCs will also target hepatocytes making it doubtful whether this approach will work. This statement should be deleted; ii) last sentence of the Introduction; iii) line 99: ‘These results suggest that parkin expression is increased in HSCs under fibrotic conditions’. This statement is misleading: The experiments were done in liver homogenates and suggest that parkin expression is increased in the liver (may not be exclusively in HSCs); iv) first sentence of section 2.6: To delineate whether TGF-b-mediated mitophagy can affect profibrotic reaction of liver”: again, misleading as the authors move on to look for gene expression in HSCs.

For all these 4 reasons, it is far from clear or even unlikely that it is a suitable target for liver fibrosis and the authors should refrain from the claim that it may be a suitable target for liver fibrosis. The only place where this could be discussed is in the Discussion, but then the authors should explicitly mention that Parkin is also upregulated in hepatocytes which may hamper its application as a target for treatment of liver fibrosis.

Figure 2 is important for the observed effects of HSC activation on Parkin expression, but some controls are missing. Panel A: markers for primary hepatocytes and primary HSCs are missing. Also, it is not clear whether these are activated or quiescent HSCs. This should be described and appropriate markers for either quiescent or activated markers should be shown. Panel B: it is questionable how presumably activated HSCs (that have lost their lipid droplets) can be isolated using the described method (which makes use of their low density due to the presence lipid droplets). Hence, again it is important to show the appropriate markers for activated HSCs. Panel D versus E: how do the authors explain the large difference in Parkin expression under the same conditions (2 ng TGF-b/ml for 12h) in Panel D and E? this should be mentioned and discussed in the text.

Figure 3: the effect of TGF-b on LC3BII expression is quite subtle, but important for the conclusions of the manuscript (the apparent paradox). These results should be quantified, e.g. by scanning of the blots and taking the mean of three different blots. Same is true for Fig.4A and 4D. It is standard to quantify fluorescent images by analysing e.g. 100 cells from 3 different experiments.

In section 2.5 (and in the Abstract: the apparent paradox) the authors state that “These results indicate that TGF-b stimulation promotes mitophagy but does not affect mitochondrial function in HSCs”. Can the authors exclude the possibility that TGF-b-induced defective mitochondria are cleared by mitophagy? The positive control experiments (rotenone) may be simply too strong for mitophagy to clear all those damaged mitochondria).

There are quite a number of papers already addressing the role of mitophagy and Parkin in HSC activation:

doi: 10.2131/jts.42.461

doi: 10.1016/j.ecoenv.2018.08.050

doi: 10.1038/s12276-018-0199-6

doi: 10.1515/med-2021-0394

doi: 10.1038/s12276-022-00923-9

The authors should mention these publications in the Introduction and Discussion and make clear how their results advance the field relative to those publications.

It is highly recommended to have the manuscript checked for use of fluent English language  

Fluency should be checked

Author Response

Reviewer #2 In this manuscript the authors describe the upregulation of the protein Parkin during TGF-b-mediated hepatic stellate cell (HSC) activation and investigate the molecular mechanisms. The results suggest an involvement of mitophagy during HSC activation. Paradoxically, TGF-b did not affect mitochondrial function but mitophagy inhibitors did affect HSC activation.

The authors show that Parkin is expressed both in hepatocytes and in HSCs. This manuscript does not address this but focusses on its expression in HSCs. However, on several occasions ‘HSC activation’ and ‘liver fibrosis’ seem to be used almost interchangeably. For example: i) the last sentence of the Abstract: since Parkin-mediated mitophagy is also likely involved in hepatocyte function, targeting parkin in HSCs will also target hepatocytes making it doubtful whether this approach will work. This statement should be deleted; ii) last sentence of the Introduction; iii) line 99: ‘These results suggest that parkin expression is increased in HSCs under fibrotic conditions’. This statement is misleading: The experiments were done in liver homogenates and suggest that parkin expression is increased in the liver (may not be exclusively in HSCs); iv) first sentence of section 2.6: To delineate whether TGF-b-mediated mitophagy can affect profibrotic reaction of liver”: again, misleading as the authors move on to look for gene expression in HSCs.

Answer: The authors appreciate the reviewer’s valuable comments. We considered parkin in HSCs would have a greater impact on liver biology based on the result from abundant parkin expression in HSCs as in compared to in hepatocytes, and the fact that the number of HSCs incremented with HSC activation during fibrosis, whereas that of hepatocytes declined [11]. Moreover, we carried out additional experiments for parkin expression by TGF-b treatment in HepG2 cells and found that parkin expression was not changed. This was discussed in the revised manuscript.  

 For all these 4 reasons, it is far from clear or even unlikely that it is a suitable target for liver fibrosis and the authors should refrain from the claim that it may be a suitable target for liver fibrosis. The only place where this could be discussed is in the Discussion, but then the authors should explicitly mention that Parkin is also upregulated in hepatocytes which may hamper its application as a target for treatment of liver fibrosis.

Answer: As the reviewer suggested, we additionally examined parkin expression in HepG2 cells and found that parkin expression was not affected by TGF-b treatment as we showed above. Hence, we focused the role of parkin induced by TGF-b in HSCs, not in hepatocytes.   

Figure 2 is important for the observed effects of HSC activation on Parkin expression, but some controls are missing. Panel A: markers for primary hepatocytes and primary HSCs are missing. Also, it is not clear whether these are activated or quiescent HSCs. This should be described and appropriate markers for either quiescent or activated markers should be shown. Panel B: it is questionable how presumably activated HSCs (that have lost their lipid droplets) can be isolated using the described method (which makes use of their low density due to the presence lipid droplets). Hence, again it is important to show the appropriate markers for activated HSCs. Panel D versus E: how do the authors explain the large difference in Parkin expression under the same conditions (2 ng TGF-b/ml for 12h) in Panel D and E? this should be mentioned and discussed in the text.

Answer: We agree with reviewer’s comments. The markers for primary hepatocyte (e.g., albumin), primary HSCs (e.g., a-SMA), and for activated HSCs by CCl4 injection (e.g. a-SMA) were additionally examined and shown in Fig. 2A and 2B of the revised manuscript. We have also described the state of primary HSCs in the manuscript. The Fig. 2D was replaced with the better one. Moreover, the result was quantified, and the relative changes were statistically analyzed as shown in Fig. 2D and 2E of the revised manuscript.     

Figure 3: the effect of TGF-b on LC3BII expression is quite subtle, but important for the conclusions of the manuscript (the apparent paradox). These results should be quantified, e.g. by scanning of the blots and taking the mean of three different blots. Same is true for Fig.4A and 4D. It is standard to quantify fluorescent images by analysing e.g. 100 cells from 3 different experiments.

Answer: According to the reviewer’s comment, LC3BII expression and merged dots of LC3B and mitochondria, representing mitophagy, were quantified and statically analyzed.

In section 2.5 (and in the Abstract: the apparent paradox) the authors state that “These results indicate that TGF-b stimulation promotes mitophagy but does not affect mitochondrial function in HSCs”. Can the authors exclude the possibility that TGF-b-induced defective mitochondria are cleared by mitophagy? The positive control experiments (rotenone) may be simply too strong for mitophagy to clear all those damaged mitochondria).

Answer: Thank you for your valuable comments. We mentioned that TGF-b stimulation promotes mitophagy but does not affect mitochondrial function in HSCs in section 2.5. We also imply that this might be derived from possibility that TGF-b-induced defective mitochondria are cleared by mitophagy. Further studies are necessary to explain this phenomenon based on the report that parkin-mediated mitophagy maintains the balance of mitochondrial dynamics. We mentioned and discussed these in the revised manuscript.

There are quite a number of papers already addressing the role of mitophagy and Parkin in HSC activation: doi: 10.2131/jts.42.461, doi: 10.1016/j.ecoenv.2018.08.050, doi: 10.1038/s12276-018-0199-6, doi: 10.1515/med-2021-0394, doi: 10.1038/s12276-022-00923-9

The authors should mention these publications in the Introduction and Discussion and make clear how their results advance the field relative to those publications.

Answer: The authors thank the reviewer for the helpful comment. As the reviewer suggested, the papers addressing the role of mitophagy and parkin were cited and discussed.

It is highly recommended to have the manuscript checked for use of fluent English language.

Answer: Our manuscript was edited by English-editing service and certification was attached.

Round 2

Reviewer 1 Report

The presentation of results is fine. However, I would suggest the authors to consider that the increased expression of aSMA is not the same as fibrosis, it is just a pathway in fibrogenesis. To establish fibrosis, excessive extracellular matrix deposition must be shown. Also in figure 7, a homodimer of either smad2 or smad3 might be rather than indicating as a monomer of smad2/3.

-

Author Response

Reviewer #1 The presentation of results is fine. However, I would suggest the authors to consider that the increased expression of aSMA is not the same as fibrosis, it is just a pathway in fibrogenesis. To establish fibrosis, excessive extracellular matrix deposition must be shown. Also in figure 7, a homodimer of either smad2 or smad3 might be rather than indicating as a monomer of smad2/3

Answer: Authors appreciate the reviewer for meticulous reading of our manuscript. We agree reviewer’s comments and edited MS as suggested. Moreover, we have modified Figure 7 in the revised manuscript.